# Black Carbon Emissions and Associated Health Impacts of Gas Flaring in the United States

**Chen Chen** [1] 🅾, **David C. McCabe** [2], **Lesley E. Fleischman** [2] **and Daniel S. Cohan** [1,*] 🅾

1 Department of Civil and Environmental Engineering, Rice University, Houston, TX 77005, USA; cc143@rice.edu
2 Clean Air Task Force, Boston, MA 02109, USA; dmccabe@catf.us (D.C.M.); lfleischman@catf.us (L.E.F.)
\* Correspondence: cohan@rice.edu

**Abstract:** Gas flaring from oil and gas fields is a significant source of black carbon (BC) emissions, a component of particulate matter that damages health and warms the climate. Observations from the Visible Infrared Imaging Radiometer Suite (VIIRS) satellite instrument indicate that approximately 17.2 billion cubic meters (bcm) of gas was flared from upstream oil and gas operations in the United States in 2019. Based on an emissions factor equation that accounts for the higher heating value of the gas, that corresponded to nearly 16,000 tons of BC emitted, though estimates vary widely across published emissions factors. In this study, we used three reduced-form air quality and health effect models to estimate the health impacts from the flaring-emitted BC particulate matter in the United States. The three models—EASIUR, AP3, and InMAP—predict 26, 48, and 53 premature deaths, respectively, in 2019. The mortality range expands from 5 to 360 deaths annually if alternative emission factors are used. This study shows that reduced-form models can be useful to estimate the impacts of numerous dispersed emissions sources such as flares, and that further research is needed to better quantify BC emissions factors from flares.

**Keywords:** flaring; methane; reduced-form models; black carbon; particulate matter; emissions; oil and gas; mortality; morbidity; health

## 1. Introduction

Flaring is commonly used at oil fields that lack the infrastructure to capture the associated gas that oil wells typically produce. Flaring is favored over direct venting because it converts methane ($CH_4$) and volatile organic compounds (VOCs) present in the associated gas to carbon dioxide ($CO_2$), a less potent greenhouse gas without direct air quality impacts [1,2]. However, incomplete combustion in flares emits black carbon (BC) particulate matter (PM), which harms health and warms the climate [3,4].

The World Bank estimated based on satellite measurements that 17.3 billion cubic meters (bcm) of gas were flared in the United States in 2019 based on satellite measurements of flaring [5], ranking it among the top three flaring countries for 2019 and corresponding to roughly 1% of US gas production. By comparison, the US Energy Information Administration (EIA) estimated based on state-reported data that 15.3 bcm was vented and flared in 2019 [6]. Venting and flaring grew as oil and gas production outpaced the growth of gas gathering and pipeline infrastructure [6]. North Dakota and Texas account for a combined 85% of reported US gas vented and flared, driven by the rapid growth of crude oil production in the Bakken formation in North Dakota and the Permian Basin and Eagle Ford shale areas in Texas [6]. In November 2021, the US Environmental Protection Agency (EPA) proposed new regulations for methane emissions from the oil and natural gas industry [7], which would allow flaring to continue to avoid direct methane emissions.

Estimates of BC emissions factors for gas flaring vary widely [8–17], since emission factors are mainly derived from small-scale laboratory experiments and limited field

observations [12]. Recent studies used aircraft [14–16] or ground-based optical methods [17] to measure BC emissions from flares in various locations. Some of the studies give a range of emission factors for BC [8,15–17], while others estimate only mean emission factors across flares [9–14]. We subsequently show details of these estimates in Table 1.

**Table 1.** Black carbon (BC) emission factors from gas flaring in previous studies.

| Emission Factor (g/m$^3$) | Source |
| --- | --- |
| 0.13–0.28 | Weyant et al. (2016) [15] |
| 0.51 | McEwen and Johnson (2012) [12] |
| 0.57 | Schwarz et al. (2015) [14] |
| 0.85 | US Environmental Protection Agency (2009) [10] |
| 0.9–6.4 | US Environmental Protection Agency (1995) [8] |
| 1.6 | Stohl et al. (2013) [13]; GAINS [11] |
| 1.83 | Conrad and Johnson (2017) [17] |
| 2.56 | Canadian Association of Petroleum Producers (CAPP) (2007) [9] |
| 0.194–4.782 | Böttcher et al. (2021) (HHV dependent) [18] |

Two studies found that BC emissions factors increase roughly linearly with the heat content of the associated gas, although the relationships remain uncertain [12,17]. The heat content of flared gas varies widely across oil and gas fields [18]. Thus, applying a single emission factor to all flares could misrepresent the BC emissions.

Several studies quantified the human health impacts of PM emissions from gas flares [19–25]. Anejionu et al. (2015) [19] and Nwosisi et al. (2021) [20] found that flaring poses a substantial threat to human health, such as respiratory and dermal diseases in the Niger Delta. Motte et al. (2021) [21] computed health impacts of PM and hydrocarbons emitted from flares globally based on emission factors and concentration-response functions, but their country-level analysis did not account for the locations of flares within countries. They showed that the vast majority of direct air pollution health impacts from flares come from PM rather than gas-phase air pollutants. Cushing et al. found in a 2020 study that proximity to flares was associated with preterm birth, shorter gestation, and lower birth weight [22], and in a subsequent study estimated that over half a million Americans live within 5 km of a flare [23]. Mirrezaei and Orkomi (2020) [24] computed health effects of air toxics but not PM from flares in Iran. Willis et al. (2020) [25] observed elevated rates of pediatric asthma hospitalizations in communities near natural gas production sites in Texas, but did not find consistent associations with flaring volumes.

Health impacts of flaring emissions are difficult to quantify via morbidity and mortality data, since those emissions are small relative to sources such as vehicles. Three-dimensional modeling can simulate health impacts across broader regions but requires computationally intensive models to represent meteorology, photochemistry, and health effects. In contrast, reduced-form models enable more computationally efficient analysis by precomputing the sensitivity of air quality and health to changes in emissions from any location within a modeling domain. Recent studies found that reduced-form models simulate health impacts consistent with each other and with three-dimensional air quality models over the United States [26–28].

In this study, we use three reduced-form air quality models, the Estimating Air pollution Social Impact Using Regression (EASIUR) model [29], the Air Pollution Emission Experiments and Policy analysis Version 3 (AP3) model [30], and the Intervention Model for Air Pollution (InMAP) [31], to quantify the morbidity and mortality impacts of all observed flares at upstream oil and gas operations in the United States in 2019. BC emission rates are computed for each flare based on the volume of flared gas and an emission factor that accounts for the higher heating value of gas in each region. We test the sensitivity of results to alternative BC emission factors and to the choice of reduced-form model.

## 2. Materials and Methods

### 2.1. Flared Gas Volume

We adopted data from the National Oceanic and Atmospheric Administration (NOAA) Visible Infrared Imaging Radiometer Suite (VIIRS) satellite instrument, as analyzed using the methodology of [32], to identify flaring sites and estimate the volume of gas flared at each site using a linear calibration method based on radiant heat. Among over 7000 identified flare sites globally, half of the flared gas volume is concentrated at fewer than 400 flares. The flared volume data are available from 2012 to 2020 ( https://eogdata.mines.edu/download_global_flare.html, accessed 6 August 2021).

Elvidge et al. (2016) [32] categorized each site as "upstream" (oil and gas production sites) or "downstream" (refineries and other processing sites). Over 97% of the US flared gas volume in 2019 came from upstream flares. For simplicity, we consider only upstream flares in this study.

### 2.2. Black Carbon Emission Factors

We used the Böttcher et al. [18] heat content-dependent emissions factor for BC from associated gas for our central case, and used results from other published emissions factors (which are not heat-content dependent) for sensitivity analyses.

For the central case, we adopted Equations (1) and (2) from Böttcher to estimate the BC emissions factor ($EF_{BC}$, in g/m$^3$ ) for each flare as a nonlinear function of the heat content (higher heating value, HHV, in MJ/m$^3$) of the gas:

$$EF_{BC} = 0.0112(\ln(HHV - 37.6))^{4.612} + 0.194, \forall HHV > 38.6 \text{ MJ/m}^3 \qquad (1)$$

$$EF_{BC} = 0.194, \forall HHV \leq 38.6 \text{ MJ/m}^3 \qquad (2)$$

The heat content of the flared gas is calculated from the gas composition reported by oil and gas production facilities to the EPA Greenhouse Gas Reporting Program (GHGRP) Subpart W [33]. Facilities are required to report $CH_4$ and $CO_2$ mole fractions and flaring $CH_4$ emissions annually from 2011 to 2020 at the county level. Thus, the weighted average $CH_4$ and $CO_2$ mole fraction from all facilities within a county can be expressed as in Equations (3) and (4):

$$\bar{X}_{CH_4} = \frac{\sum_{i=1}^{n} X_{i,CH_4} * M_{i,CO_2}}{\sum_{i=0}^{n} M_{i,CO_2}} \qquad (3)$$

$$\bar{X}_{CO_2} = \frac{\sum_{i=1}^{n} X_{i,CO_2} * M_{i,CO_2}}{\sum_{i=0}^{n} M_{i,CO_2}} \qquad (4)$$

where $n$ denotes total number of facilities within a county and $i$ denotes the facility. $X_{i,\,CH_4}$ is the mole fraction of $CH_4$, $X_{i,\,CO_2}$ is the mole fraction of $CO_2$, and $M_{i,CO_2}$ is the flaring $CO_2$ emissions reported by facility $i$.

The sum of weighted average $CH_4$ and $CO_2$ mole fractions is always smaller than one because there are other constituent species of associated gas such as ethane, propane, etc. For the nonmethane and non-$CO_2$ composition, we used the Energy Information Administration (EIA) data, listed in the Appendix A Table A1, on regional natural gas liquid (NGL) production ratio [6] to represent the remaining mole fraction. Once we obtained the full gas composition, we calculated the heat content of the flared gas from each county following Equation (5):

$$\begin{aligned} HHV_{county} = \bar{X}_{CH_4} \times HHV_{CH_4} + \bar{X}_{ethane} \times HHV_{ethane} + \bar{X}_{propane} \times HHV_{propane} \\ + \bar{X}_{butane} \times HHV_{butane} + \bar{X}_{pentane} \times HHV_{pentane} \end{aligned} \qquad (5)$$

where $\bar{X}_x$ indicates the mole fraction in associated gas of constituent X (i.e., methane, ethane, propane, butane, pentane), and $HHV_x$ indicates the heat content of constituent x. We were able to directly calculate the heat content of over 95% of flares observed by VIIRS in the US in 2019 using this method.

For upstream flares in counties that had no data reported in EPA's GHGRP, we estimated the heat content using one of the following methodologies:

1.  Average gas composition (weighted by flaring volume) in a neighboring county (applied to flares accounting for 1% of flaring volume);
2.  Average gas composition (weighted by flaring volume) in the entire basin, for counties where option 1 is inapplicable (applied to flares accounting for 1% of flaring volume);
3.  Simple average of gas composition in the entire basin for counties where options 1 and 2 are both inapplicable (applied to flares accounting for 2% of flaring volume).

For sensitivity analysis, we used the heat content-independent BC emission factors reported in other studies (Table 1).

### 2.3. Black Carbon Emissions

BC emissions from gas flaring in the United States in 2019 were estimated by multiplying the BC emission factors by the volume of gas flared at each site *i*, as shown in Equation (6):

$$Mass\ of\ BC\ =\ \sum_{i=1}^{n} EF_{BC} \times V \tag{6}$$

We assume a ground-level stack height based on the EPA's Air Pollution Control Cost Estimation Spreadsheet for Elevated Flares, which shows that the typical heights for flares are lower than 15 m [34]. Sensitivity analysis with EASIUR and AP3 indicates that assuming a stack height of 100–200 m would reduce impact estimates by approximately 5–20%. We use annual data for flared gas volumes, which are more robust than monthly data because the Colorado School of Mines Payne Institute Earth Observation Group (EOG) develops them based on clear-sky observations throughout the year rather than as a sum of monthly estimates (personal communication with EOG, 6 August 2021).

### 2.4. Reduced-Form Models

Given the expense involved in running three-dimensional atmospheric models, a number of reduced-form models were developed and made available to the research community in recent years. These tools were used to estimate health impacts from a number of diverse sources of pollution in the United States. We briefly describe the three models used in this study, with references to full descriptions of these models.

EASIUR is an online model which estimates the $PM_{2.5}$-caused mortality directly from emissions of primary $PM_{2.5}$ (prPM$_{2.5}$) and three precursor gases—sulfur dioxide ($SO_2$), nitrogen oxides ($NO_x$), and ammonia ($NH_3$) [29] (Table 2). EASIUR derives health impacts from regressions on a dataset consisting of small emissions perturbations at 100 sample locations using the CAMx photochemical model [35]. The base meteorology and emissions are simulated with a 36 km resolution based on Year 2005 conditions. The EASIUR model reports the marginal damages ($/t) for the four emitted species at three stack heights during four seasons and an annual average. The default concentration–response function (CRF) used in EASIUR is taken from a cohort study conducted by the American Cancer Society [36]. EASIUR presents marginal health damages as a look-up table without providing the air quality changes that led to those health damages.

AP3 is the updated version of the APEEP model, which has been widely used for policy analyses related to air pollution in the United States (https://public.tepper.cmu.edu/nmuller/APModel.aspx, accessed 4 October 2021). AP3 takes county-level emissions of $PM_{2.5}$ and its precursors and simulates atmospheric transport, chemical transformation, and deposition across the contiguous United States. AP3 links emissions in source counties (s) to ambient $PM_{2.5}$ concentration changes in receptor counties (r) via a source-receptor matrix that was precomputed by a modified Gaussian plume model [37,38]. In this study, we assume flaring emissions are released at ground level. To estimate the corresponding health effects, AP3 uses a CRF relating the average annual $PM_{2.5}$ concentration to annual mortality for adults older than 30 years old [36] and infants younger than one year old [39].

**Table 2.** Intercomparison of three reduced-form models, including resolution, input emissions, and outputs.

| Reduced-Form Models | Resolution | Input Emissions | Outputs | Reference |
|---|---|---|---|---|
| Estimating Air Pollution Social Impacts Using Regression (EASIUR) | 36 km | Primary $PM_{2.5}$ *, $SO_2$, $NO_x$, and $NH_3$ | Marginal damage ($/ton) | Heo et al., 2016 [29] |
| Air Pollution Emission Experiment and Policy Analysis Model (AP3) | US Counties | Primary $PM_{2.5}$ *, $SO_2$, $NO_x$, $NH_3$, and VOC | $PM_{2.5}$ ($\mu g/m^3$), mortality per county | Muller, 2014 [30] |
| Intervention Model for Air Pollution (InMAP) | 1–288 km | Primary $PM_{2.5}$ *, $SO_2$, $NO_x$, $NH_3$, and VOC | $PM_{2.5}$ ($\mu g/m^3$), mortality per grid cell | Tessum, Hill, et al., 2017 [31] |

* Focus of this study.

The InMAP reduced-form air quality model uses spatially resolved annual average photochemical relationships derived from a state-of-the-science WRF-Chem chemical transport model [40] to simulate the formation and transport of primary and secondary $PM_{2.5}$ [31]. InMAP also developed a S–R matrix to create spatially explicit estimates of ambient $PM_{2.5}$ concentration differences caused by primary $PM_{2.5}$ emissions from flaring. InMAP runs at a varying spatial resolution with cell length ranging from 1–288 km based on population density. InMAP also estimates premature mortalities with its internally embedded CRF using relative risks from [36,41].

*2.5. Health Impacts*

We apply all three models to compute health impacts per ton of emissions from each flaring site. EASIUR outputs only monetized mortality impacts and only on a domain-wide aggregated basis. We divide the monetized outputs by the model's value of a statistical life (VSL) [29] to convert them into excess mortalities. The other models indicate the county (AP3) or grid cell (InMAP) where impacts occur.

To further compare the performance of AP3 and InMAP, we aggregated results to the county- and state-level. AP3 provides results on a county-level basis, thus aggregation to the state level is straightforward. For InMAP, health impacts in grid cells that straddle state or county boundaries were divided equally among those states or counties.

We also estimated the morbidity incidences by taking the $PM_{2.5}$ concentration surfaces generated by AP3 and InMAP into EPA's Environmental Benefits Mapping and Analysis Program-Community Edition (BenMAP-CE) model [42]. We selected several of the health endpoints from BenMAP-CE to provide examples of nonmortality health impacts from flaring. Those endpoints are taken from the EPA Standard Health Functions (2021), which reflects recent epidemiological literature from EPA's Integrated Science Assessments for $PM_{2.5}$ and Ozone (BenMAP-CE Release Notes 2021). We focus on 8 of the 64 $PM_{2.5}$-related morbidity endpoints represented in EPA's analysis: acute myocardial infarction [43], ER visits due to respiratory distress [44], Alzheimer's disease [45], asthma [46], lung cancer [47], stroke [48], and work loss days [49]. All morbidity estimation includes a 95% confidence interval (2.5–97.5%). We do not monetize morbidity outcomes due to a lack of consensus cost per endpoint estimate.

## 3. Results

### 3.1. BC Emissions from Flaring

Figure 1 shows the spatial distribution of flare sites identified by VIIRS, with the colors indicating BC emissions per site in 2019. The color of points indicates the annual BC emissions per site (red: high; green: low) [32]. VIIRS identified 2652 flare sites spanning 22 states and offshore locations, with a total volume of 17.21 bcm of gas flared. North Dakota and Texas, as the two largest oil and gas producing states, had the most flaring observed in 2019, as shown in the enlarged maps in Figure 1.

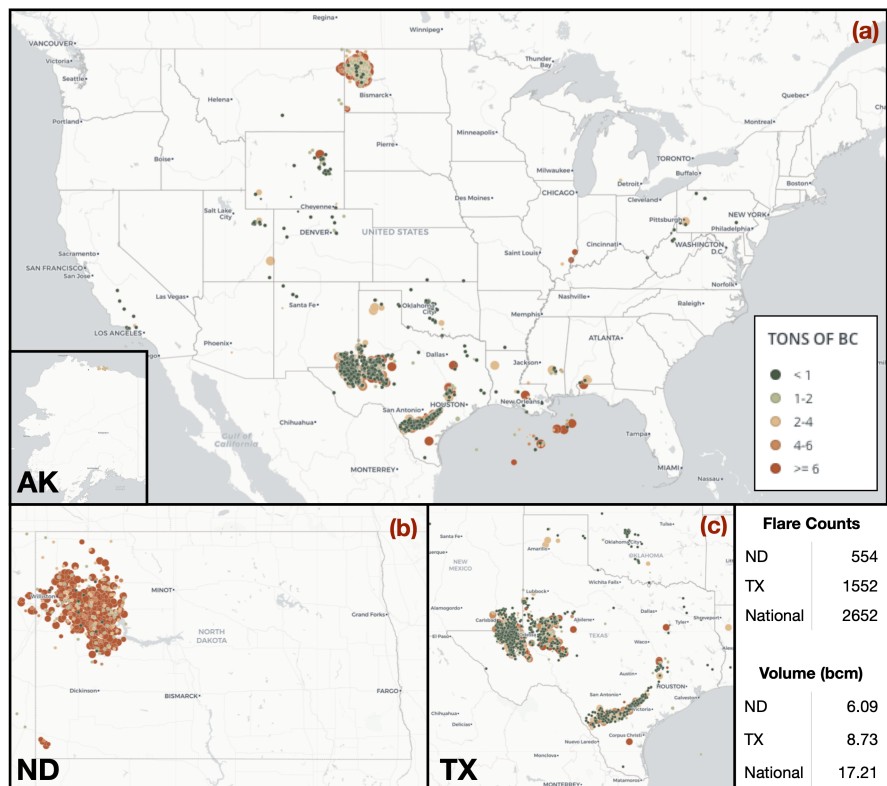

**Figure 1.** Locations of gas flaring sites in (**a**) contiguous US (plus Alaska), (**b**) North Dakota, and (**c**) Texas. Color and size of circles denote annual tons of BC emitted per site in 2019 (red: high; green: low).

Based on the flaring data, 22 states in the United States plus the offshore regions flared 17.21 bcm gas in 2019. Figure 2 shows the flared gas volume by state, led by Texas (8.73 bcm), North Dakota (6.09 bcm), and New Mexico (1.38 bcm). Over 95% of the volumes came from the Permian (TX, NM), Williston (ND, MT), and Western Gulf (TX) basins (See Figure A1 in the Appendix A).

BC emissions factors vary widely across sites, ranging from 0.19 to 4.78 g/m$^3$, with an average of 0.76 g/m$^3$, varying with differences in gas composition (Figure 3). Among states with more than 10 flaring sites, Montana and North Dakota have the highest emission factors (1.87 and 1.74 g/m$^3$) due to the high HHVs (>55 MJ/m$^3$) of the gas flared there. The weighted average of emission factors in Texas is 0.48 g/m$^3$, since its HHV is lower than the national average.

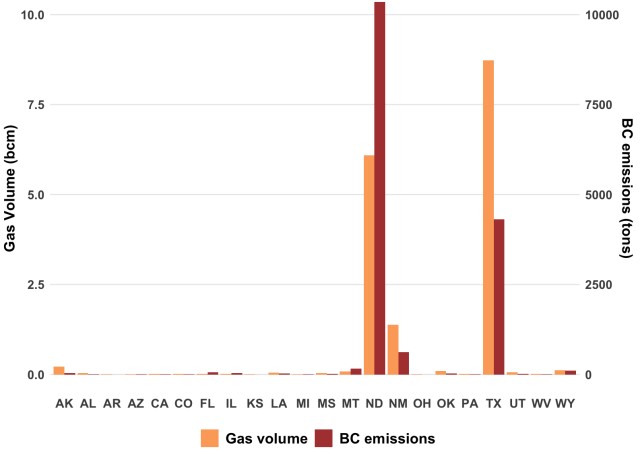

**Figure 2.** Flared gas volume (yellow bars) and BC emissions (red bars) by state in 2019.

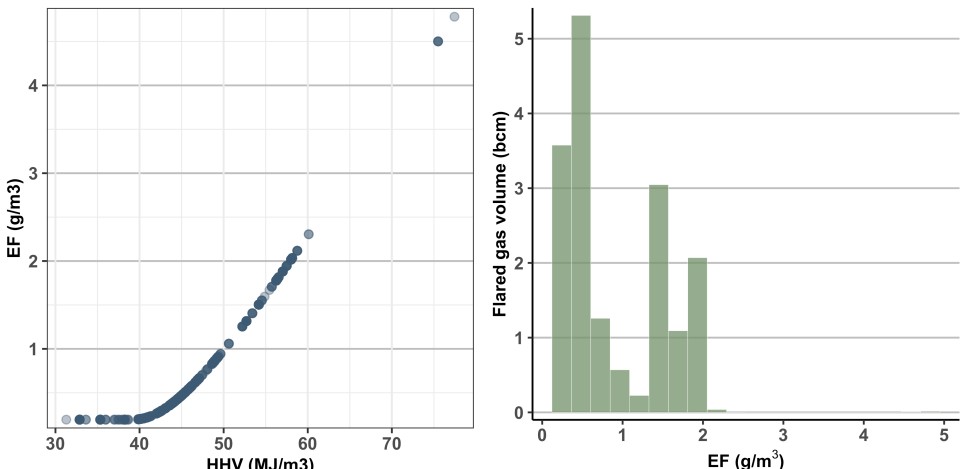

**Figure 3.** BC emission factors estimated for all flares based on Böttcher et al. [18]. **Left**: relationship between HHV (MJ/m$^3$) and emissions factor (EF) (g/m$^3$), where darkness indicates number of points overlapped. **Right**: a histogram of black carbon EFs, weighted by volume of gas flared.

BC emissions from upstream flaring in the contiguous United States totaled 15,968 tons, with per-site levels ranging from 0.04 to 113.68 tons and averaging 6 tons. North Dakota and Texas led all states with 10,036 and 4317 tons, respectively, of BC emissions from flaring, combining for 89.9% of the total BC emissions from flaring nationwide (Figure 2). Two basins, Williston Basin and Permian Basin, emitted 10,519 and 3913 tons of BC, respectively, corresponding to 90.4% of total emissions.

### 3.2. Air Quality

EASIUR outputs only domain-wide mortality impacts, leaving spatial patterns of ambient air quality impacts unknown. Thus, we focus our air quality analysis on AP3 and InMAP, which provide the PM$_{2.5}$ concentration changes simulated by its underlying air quality model; thus, the air quality impacts from flaring cannot be estimated from the EASIUR model.

By contrast, both AP3 and InMAP provide spatially resolved changes in ambient PM$_{2.5}$ resulting from emissions. Among 3109 counties, AP3 simulated the greatest PM$_{2.5}$ concentration increases in eight North Dakota counties, followed by 10 Texas counties and one Montana county. Four core oil and gas producing counties—McKenzie, Mountrail, Williams, and Dunn in North Dakota—had modeled PM$_{2.5}$ increases of more than 0.84 μg/m$^3$. Due to their low baseline PM$_{2.5}$ concentrations, flaring was responsible for 16–25% of total PM$_{2.5}$ in those counties.

Since the InMAP model provides outputs on grid cells with resolution ranging from 1–288 km, we spatially joined all grids to state polygons and labeled each grid cell by state. If a grid cell lays on the state boundary, it would be assigned with multiple states and all states would incorporate the ambient PM$_{2.5}$ change from this cell. Modelled PM$_{2.5}$ increases were highest in a cell along the North Dakota–Montana border (0.26 μg/m$^3$), followed by a cell in Texas (0.25 μg/m$^3$).

To compare with AP3's results, we also aggregated InMAP results to the county-level. InMAP simulated smaller peak PM$_{2.5}$ impacts than AP3, with the largest PM$_{2.5}$ impact being 0.26 μg/m$^3$ in Williams County, ND and impacts of at least 0.10 μg/m$^3$ in 45 additional counties.

### 3.3. Health Impacts

We applied all three models to simulate health impacts from our base case BC emissions estimates for 2019. EASIUR simulates that those emissions caused 53 deaths, including 32 deaths from North Dakota emissions and 16 from Texas emissions, but does not indicate

where the deaths occurred (see Figure A2 in the Appendix A). By contrast, AP3 and InMAP provide source-receptor matrices that allow us to examine the locations of both emissions and health impacts. Figure 4 maps the receptor counties of mortality impacts based on AP3 and InMAP simulations. AP3 estimates a total of 48 excess deaths, while InMAP predicts only 26 deaths nationwide.

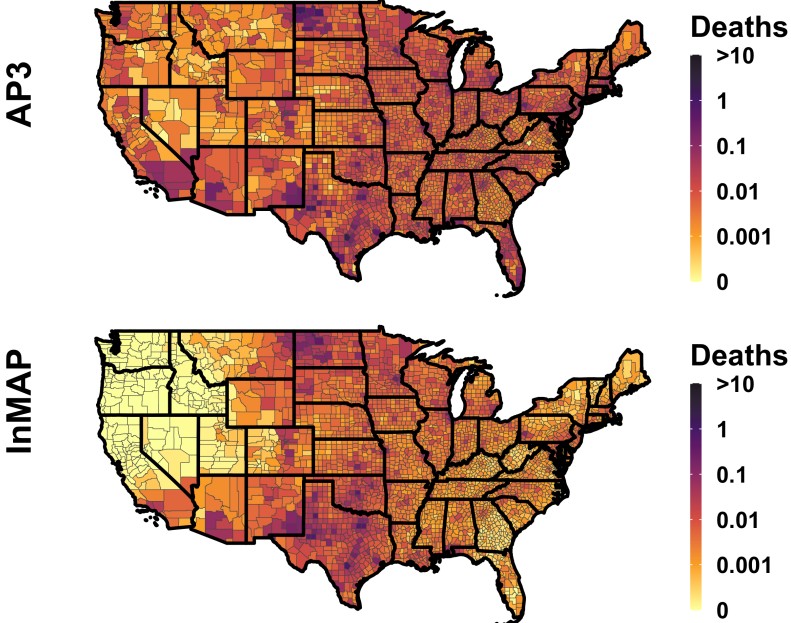

**Figure 4.** Spatial distribution of flaring-caused mortality in contiguous US as simulated by the AP3 (**top**) and InMAP (**bottom**) models, plotted by receptor county.

The receptor counties of health impacts simulated by AP3 and InMAP are plotted in Figure 5a. Assuming a zero intercept, the linear correlation between AP3 and InMAP results on a county-level basis has an $R^2$ of 0.62. A majority of counties have nearly zero mortality, and the county with highest estimation of flaring-caused deaths from both models is Bexar County, Texas, due to its large population and proximity to the Western Gulf Basin (Eagle Ford shale).

Figure 5b compares the receptor locations of AP3 and InMAP mortality estimates on a state-level basis. AP3 simulates that gas flaring causes 13 deaths in Texas, which is the highest among all states, and another four deaths in North Dakota. Ten other states (IL, FL, MO, OH, MI, MN, OK, PA, WI, IN) are modeled to have at least one death despite very little flaring within those states, due to transport of emissions from other states. InMAP, on the other side, predicts only 10 deaths occurring in Texas and two in North Dakota.

In addition to mortality, flaring-caused morbidity was estimated using BenMAP-CE, which translated the air pollution surfaces generated by AP3 and InMAP to selected morbidity endpoints (Table 3). AP3 estimates nationwide impacts of 1.1 acute myocardial infarctions, 51.6 emergency room visits for respiratory cases, 23.5 cases of Alzheimer's disease, 178.2 pediatric asthma onset, 4.2 lung cancers, and 8,481 lost workdays, whereas InMAP estimates less than half as many incidents for each outcome.

### 3.4. Sensitivity to Emission Factors

Given the uncertainty across published estimates of emission factors, we conducted a sensitivity analysis by applying alternative emission factors from Table 1 in place of our base-case estimate from [18] (Equation (1) and (2)) to understand how they influence mortality estimates. In Figure 6, all three models establish a linear correlation between the emission factors and predicted mortality. AP3 and EASIUR show highly consistent mortality estimates as the emission factor changes, while InMAP predicts a lower mortality value.

Under different emission factors, the flaring-caused deaths have fallen in a wide range between 5 and 368 annually, with a median of 51. Thus, the models are highly sensitive to the emission factor used to estimate BC emissions. All mortality results predicted from the three models under the alternative emission factors can be found in the Appendix A (Table A2).

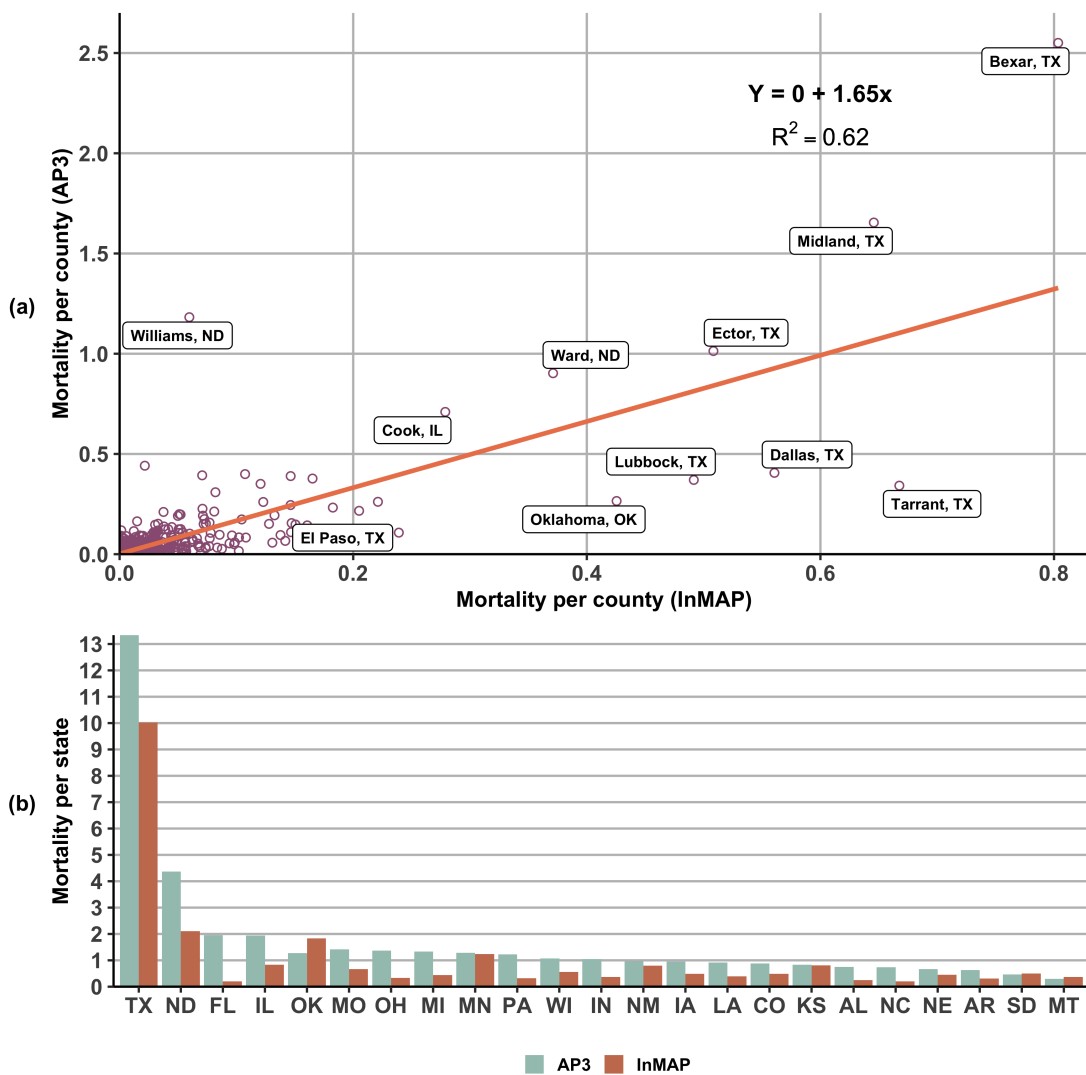

**Figure 5.** AP3 and InMAP simulations of flaring induced mortalities based on location of impacted population, plotted by (**a**) county and (**b**) state. Line in (**a**) assumes a linear fit and zero intercept.

**Table 3.** BenMAP-CE estimates of morbidity caused by US flaring BC emissions in 2019, based on $PM_{2.5}$ contributions modeled by AP3 and InMAP and dose-response relationships from cited sources.

| Endpoint | Source | Morbidity (AP3) (Incidents per Year) | Morbidity (InMAP) (incidents per Year) |
|---|---|---|---|
| Acute myocardial infarction | Zanobetti et al (2009) | 1.1 (95% CI: 0.5, 1.7) | 0.4 (95% CI: 0.2, 0.7) |
| ER visits (Respiratory) | Krall et al. (2016) | 51.6 (95% CI: 8.5, 93.1) | 20.4 (95% CI: 3.4, 36.8) |
| Alzheimer's disease | Kioumourtzoglou et al. (2016) | 23.5 (95% CI: 17.6, 29.3) | 12.2 (95% CI: 9.1, 15.2) |
| Asthma onset | Tetreault et al. (2016) | 178.2 (95% CI: 171.0, 185.2) | 72.0 (95% CI: 69.0, 74.8) |
| Lung cancer | Gharibvand et al. (2016) | 4.2 (95% CI: 1.3, 6.9) | 1.7 (95% CI: 0.51, 2.8) |
| Stroke | Kloog et al. (2012) | 3.3 (95% CI: 0.9, 5.7) | 1.4 (95% CI: 0.3, 2.3) |
| Work loss days | Ostro (1987) | 8480.6 (95% CI: 7147.7, 9764.7) | 3383.2 (95% CI: 2851.4, 3895.5) |

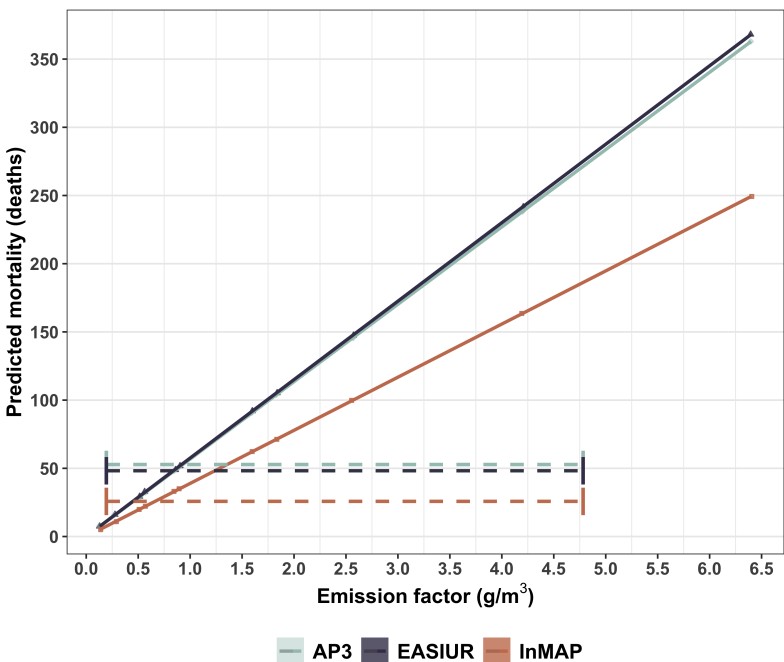

**Figure 6.** Sensitivity of US flaring BC mortality estimates to emission factor and model used. Horizontal bars reflect range of emission factors from [18].

## 4. Discussion

The flaring of natural gas from the upstream oil and natural gas industry in the United States emitted approximately 16,000 tons of BC in 2019, according to our baseline estimates. That corresponds to approximately 8% of nationwide total BC emissions, or 10% of total anthropogenic BC emissions [50]. Most emissions originated from North Dakota and Texas, since over 86% of nationwide flaring occurred in those two states. Although North Dakota flared a lower volume of gas than Texas, it emitted more BC because the heat content of North Dakota gas, and thus the BC emission factors of its flares were higher than those in Texas.

Based on three reduced-form models and our baseline emission factors (Equations (1) and (2)) [18], BC emissions from flares were estimated to be responsible for 26–53 excess premature mortalities in 2019. Based on a $8.5 million VSL, that would correspond to a monetized annual cost of $219 to $449 million. The most important source of uncertainty in these results is the uncertainty in the emission factor for BC from flaring; the choice of reduced-form model is secondary. Under the highest black carbon emissions factor, flaring could potentially cause over 360 deaths per year. Future research is needed to narrow the uncertainty in BC emission factors for flares.

At the time of this writing (December 2021), EPA proposed aggressive actions to reduce direct methane emissions from the oil and natural gas industry and is considering requirements to reduce associated gas flaring. Controlling or eliminating flaring activities requires a reduction in oil and gas production or the build-out of infrastructure to capture, transport, and process associated gas. The lack of effective regulations and policies is another culprit for routine gas flaring. Current solutions such as utilizing gas gathering pipelines, small-scale gas utilization technologies [51], or reinjecting the gas back into the ground are mature; however, they are not universally deployed due to a lack of planning, regulatory requirements, and economic incentives. To tackle the problem for the future, the World Bank and the UN Secretary-General launched the Zero Routine Flaring by 2030 (ZRF) initiative to commit governments and oil companies to end routine gas flaring in new oil fields and eliminate the existing flaring before 2030 [52]. ExxonMobil announced in December 2021 that it aims to achieve net-zero greenhouse gas emissions from its operations in the Permian Basin by 2030 [53].

Beyond their direct health impacts, BC emissions can also contribute to climate change by absorbing solar radiation in the atmosphere, influencing clouds, and reducing the surface albedo and accelerating the melting of ice and snow [4,54,55]. However, quantifying the climate effects of BC requires detailed modeling of its impacts on radiation and clouds and its deposition onto snow and ice, which is beyond the scope of this study.

A parallel objective of this study is to evaluate the consistency of three reduced-form models by comparing the mortality predicted from flaring in the US. Estimates of US deaths from flaring in 2019 were similar in AP3 (48 deaths) and EASIUR (53 deaths), and a factor of two lower in InMAP (26 deaths). Results were strongly correlated spatially for the two models with spatially resolved impacts. Consistency across the reduced-form models is at least as strong as that observed across full-form atmospheric models in intercomparison studies such as the Air Quality Model Evaluation International Initiative [56,57] and the Model Intercomparison Studies–Asia [58,59].

Overall, the reduced-form models are shown to be highly efficient in estimating flaring-induced human health impacts and to yield results that are sufficiently consistent to inform policy. Although this study lacks a comparison with a full-form chemical transport model to validate the results, previous studies [26,27] provided detailed inter-comparisons of model performance for other emission scenarios.

In an evaluation of the three reduced-form models used here, Industrial Economics, Incorporated (IEc) found that AP3 and EASIUR agreed well with the paired state-of-the-science Community Multiscale Air Quality (CMAQ) [60] and BenMAP-CE models [42] in estimating excess mortalities from scenarios involving power plants, cement kilns, pulp and paper mills, and refineries, while InMAP produced higher estimates [26]. Here, we likewise found AP3 and EASIUR to yield similar estimates, but found InMAP to yield lower estimates for flaring impacts. Gilmore et al. [27] found all three reduced-form models to yield estimates of PM responsiveness to emissions that were consistent with WRF-Chem [40], and they also produced similar value across all counties for ground-level primary $PM_{2.5}$.

The limitations of this study arise mainly from the availability of flaring data, the uncertainty in emission factors, and the US-only scope of these reduced-form models. Our use of annually aggregated estimates of flaring volumes from VIIRS precluded us from considering the seasonality of air quality responsiveness to emissions. To our knowledge, these reduced-form models have not been extended to other parts of the world, except for a recent adaptation of InMAP to China [61].

## 5. Conclusions

Although gas flaring reduces direct venting of methane and other hydrocarbons, their black carbon (BC) emissions are unhealthful and deadly. Associated mortalities in 2019 in the US likely numbered in the dozens, with uncertainty in the estimates arising more from uncertainty in the BC emission factor than from the choice of reduced-form model. Our study demonstrates that reduced-form models can be useful tools for estimating the impacts of numerous dispersed emissions sources, such as flares, and that further research is needed to improve estimates of BC emission rates from flares.

**Author Contributions:** Conceptualization, C.C., D.S.C., D.C.M. and L.E.F.; methodology, C.C., L.E.F. and D.S.C.; software, C.C.; validation, C.C., D.S.C., D.C.M. and L.E.F.; formal analysis, C.C.; investigation, C.C., D.S.C., D.C.M. and L.E.F.; resources, C.C.; data curation, C.C. and L.E.F.; writing—original draft preparation, C.C.; writing—review and editing, D.S.C., D.C.M. and L.E.F.; visualization, C.C.; supervision, D.S.C.; project administration, D.C.M. and D.S.C.; funding acquisition, D.C.M. and D.S.C. All authors have read and agreed to the published version of the manuscript.

**Funding:** This research was funded by Clean Air Task Force.

**Institutional Review Board Statement:** Not applicable.

**Informed Consent Statement:** Not applicable.

**Data Availability Statement:** The flared volume data are available from 2012 to 2020 (https://eogdata.mines.edu/download_global_flare.html (accessed 6 August 2021)).

**Acknowledgments:** The authors thank Jinhyok Heo, Nick Muller, and Christopher Tessum for making their reduced-form models open source for researchers to use.

**Conflicts of Interest:** The authors declare no conflict of interest.

## Appendix A

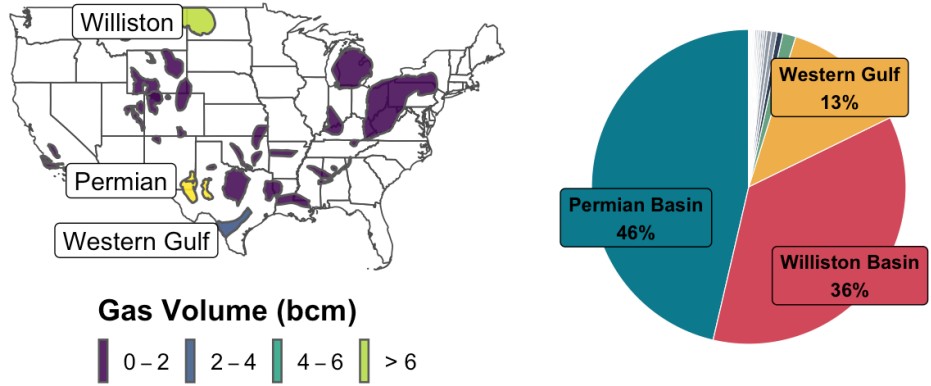

**Figure A1.** Volume of gas flared per US basin. Over 95% of US flaring in 2019 occurred in Permian, Williston, and Gulf Coast basins. Map on left shows all shale oil and natural gas basins in lower 48 states, with three leading basins labeled.

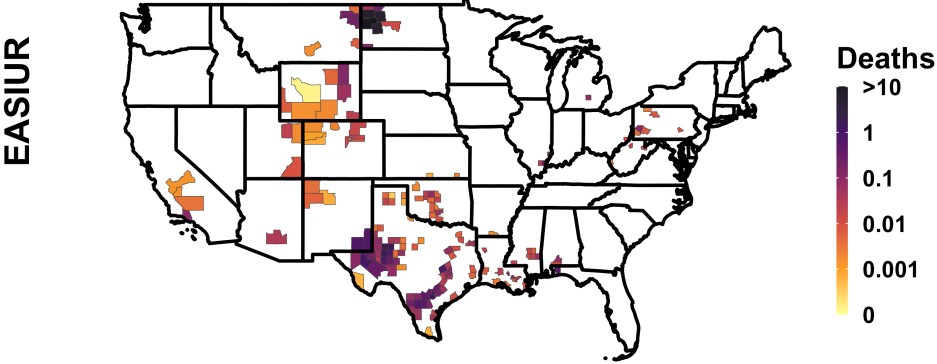

**Figure A2.** Spatial distribution of flaring-caused mortality simulated by EASIUR, plotted by emission county.

**Table A1.** Energy Information Administration (EIA) data on regional natural gas liquid (NGL) production ratio.

| Region | Area | Ethane | Propane | Butane | Isobutane | Natural Gasoline (Pentane Plus) |
|--------|------|--------|---------|--------|-----------|----------------------------------|
| PADD 1 | East Coast | 0% | 33% | 67% | 0% | 0% |
|        | Appalachian | 39% | 35% | 11% | 5% | 10% |
|        | IN, IL, & KY | 28% | 44% | 10% | 10% | 8% |
| PADD 2 | MN, WI, ND, & SD | 21% | 40% | 18% | 5% | 16% |
|        | OK, KS, & MO | 41% | 32% | 11% | 6% | 11% |
|        | LA (Gulf) | 38% | 33% | 11% | 7% | 10% |
| PADD 3 | N. LA & AR | 28% | 26% | 11% | 9% | 26% |
|        | NM | 41% | 32% | 10% | 7% | 11% |
|        | TX (Inland) | 43% | 31% | 10% | 6% | 10% |
| PADD 4 | Rocky Mountain | 25% | 37% | 14% | 7% | 16% |
| PADD 5 | West Coast | 0% | 17% | 16% | 21% | 46% |

**Table A2.** Predicted mortality from three reduced-form models using the alternate emission factors for the flaring BC.

| EF (g/m$^3$) | Source | EASIUR | AP3 | InMAP |
|---|---|---|---|---|
| 0.13 | Weyant et al. (2016) | 7 | 7 | 5 |
| 0.28 | Weyant et al. (2016) | 16 | 16 | 11 |
| 0.51 | McEwenand Johnson (2012) | 29 | 29 | 20 |
| 0.57 | Schwarz et al. (2015) | 33 | 32 | 22 |
| 0.85 | US Environmental Protection Agency (2009) | 49 | 48 | 33 |
| 0.9 | US Environmental Protection Agency (1995) | 52 | 51 | 35 |
| 1.6 | Stohl et al. (2013); GAINS | 92 | 91 | 62 |
| 1.83 | Conrad and Johnson (2016) | 105 | 104 | 71 |
| 2.5632 | CAPP (2007) | 147 | 145 | 100 |
| 4.2 | US Environmental Protection Agency (1995) | 242 | 238 | 164 |
| 6.4 | US Environmental Protection Agency (1995) | 368 | 363 | 249 |
| 0.194–4.782 | Bottcher et al. (2021) | 53 | 48 | 26 |

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
