# Peer review of "Black Carbon Emissions and Associated Health Impacts of Gas Flaring in the United States"

_atmosphere, doi:10.3390/atmos13030385_

Round 1
Reviewer 1 Report
The main idea of the paper is really interesting, but there are some suggestions for the revision:
- There are some reference lumps in the text. Please eliminate this lump. After that, please check the manuscript thoroughly and eliminate ALL the lumps. This should be done by characterizing each reference individually. This can be done by mentioning 1 or 2 phrases per reference to show how it is different from the others and why it deserves mentioning. This is not just a formalism. Having reference lumps in the text casts a serious doubt on whether the authors have really read and understood the cited sources. If you do not characterize the references individually, it matters little how they are formatted. What really matters is to have meaningful references and the requirement for individual characterization aims exactly at that. Even the small lumps leave something unsaid and reduce the quality and impact of the paper.
- The results section should make a deeper and more detailed analysis of the interesting information generated in the paper. It is suggested that the authors focusing more on the conditions and the range of applicability and the conditions in which the proposed method could not be used.
- It would be great if the authors could compare the results with other researchers work if any. The authors should compare proposed method with other methods to emphasize the advantages? This valuable information could help other researchers which working in this field of study.
As a conclusion, the paper is interesting and eventually publishable but I think that the authors should re-write it with taking into account the above remarks.
Author Response
The attached file provides our detailed response to reviewers, in which we explain how we have fully adopted the suggestions of both reviewers. We also provide versions of our manuscript that (i) show tracked changes and (ii) incorporate those changes into a clean version.

Reviewer 2 Report
Interesting article that provide good experimental information about black carbon emissions from gas flaring and associated health impact.
Two important concerns that need to be addressed:
- page 4 of 18, lines 159-160: you mentioned that you assumed flaring emissions are released at ground level. What is the accuracy of the three reduced-form models based on this assumption.
- page 8 of 18, 3.3 health impacts: Do the AP3 and InMAP simulations consider confounder effects into the mortality estimation (ex. age, gender, comorbidity, smoking status, etc.)? If not, how will this affect the results?
- Minor comments: (1) page 5 of 18, line 197: it should be 2625 flare sites and not 2652 flare sites; (2) at the end of Figure 1, write (red: high; green: low); (3) page 8 of 18, line 247: it should be (Figure 4) and not (igure 4); (4) Figure 1 should have (a), (b), (c) labelled on each figure, while Figure 5 should have (a) and (b) labelled on each figure.
Author Response

(The authors gave the same response as above.)

Round 2
Reviewer 2 Report
All suggestions are incorporated in the revised version.